# Inhibition of Eukaryotic Translation Initiation Factor 5A (eIF5A) Hypusination Suppress p53 Translation and Alters the Association of eIF5A to the Ribosomes

**DOI:** 10.3390/ijms21134583

**Published:** 2020-06-27

**Authors:** Marianna Martella, Caterina Catalanotto, Claudio Talora, Anna La Teana, Paola Londei, Dario Benelli

**Affiliations:** 1Division of Cancer Therapeutics, The Institute of Cancer Research, London SW7 3RP, UK; 2Department of Molecular Medicine, Sapienza University of Rome, Viale Regina Elena 291/324, 00161 Rome, Italy; caterina.catalanotto@uniroma1.it (C.C.); claudio.talora@uniroma1.it (C.T.); paola.londei@uniroma1.it (P.L.); 3Department of Life and Environmental Science, Polytechnic University of Marche, 60131 Ancona, Italy; a.lateana@univpm.it

**Keywords:** colon cancer cell lines, eIF5A, GC7, p53

## Abstract

The eukaryotic translation initiation factor 5A (eIF5A) is an essential protein for the viability of the cells whose proposed function is to prevent the stalling of the ribosomes during translation elongation. eIF5A activity requires a unique and functionally essential post-translational modification, the change of a lysine to hypusine. eIF5A is recognized as a promoter of cell proliferation, but it has also been suggested to induce apoptosis. To date, the precise molecular mechanism through which eIF5A affects these processes remains elusive. In the present study, we explored whether eIF5A is involved in controlling the stress-induced expression of the key cellular regulator p53. Our results show that treatment of HCT-116 colon cancer cells with the deoxyhypusine (DHS) inhibitor N1-guanyl-1,7-diamineheptane (GC7) caused both inhibition of eIF5A hypusination and a significant reduction of p53 expression in UV-treated cells, and that eIF5A controls p53 expression at the level of protein synthesis. Furthermore, we show that treatment with GC7 followed by UV-induced stress counteracts the pro-apoptotic process triggered by p53 up-regulation. More in general, the importance of eIF5A in the cellular stress response is illustrated by the finding that exposure to UV light promotes the binding of eIF5A to the ribosomes, whereas UV treatment complemented by the presence of GC7 inhibits such binding, allowing a decrease of de novo synthesis of p53 protein.

## 1. Introduction

Protein synthesis is one of the molecular processes that allow the cells to respond quickly to the changing environment. It occurs in four principal steps: initiation, elongation, termination and ribosome recycling. For many decades, protein synthesis was considered a supporting actor of gene expression, acting passively on the amount of mRNAs present in the cells at any given time. However, growing proteomic and transcriptomic data show that mRNA and protein levels are not so strictly correlated [1] and that translation changes of certain mRNAs can have physiological roles on distinct cellular processes [2]. The mechanism of eukaryotic translation is highly complex [3] and regulated at multiple levels [4]. Sulima et al. [5] have demonstrated the connection between certain components of the translation machinery and specific aspects of cell physiology, allowing the expression of certain genes rather than others. The eukaryotic translation initiation factor 5A (eIF5A) is one of the elements of translation machinery. It is a protein of 16–18 kDa, highly conserved from archaea to humans [6]. It is the only cellular protein containing a uniquely modified lysine, termed hypusine, which is essential for protein function [7,8]. In eukaryotes, IF5A is encoded by two different genes (EIF5A1 and EIF5A2), expressing two proteins that in Homo sapiens are 84% identical and are differentially transcribed among tissues and in some tumor cell lines [9]. Hereafter, we will refer to the protein eIF5A1 as eIF5A. eIF5A is constitutively expressed in mammalian cells [10] and is essential for normal mammalian development and is abundant in proliferating cells [11,12] such as in human cancer tissues [10,13]. eIF5A was initially isolated from the high-salt washes of reticulocyte lysate ribosomes with other initiation factors and shown to stimulate the formation of the first peptide bond [13,14]. For these reasons, eIF5A was initially classified as a translation initiation factor. However, subsequent work based on genetic studies and ribosome profiling analysis in yeast and human cell lines redefined its role at the level of elongation [15,16] and termination phases [17] of protein synthesis. The residue hypusine appears to have an essential role, augmenting the rate of peptide bond formation in the ribosomal catalytic center [15,16]. Furthermore, eIF5A has also been implicated in nuclear export of HIV-1 Rev [18], mRNA degradation [19], cell differentiation [20], nucleocytoplasmic transport [18], transcription [21] and viral replication [22]. The factor has also been proposed to be involved in the regulation of apoptosis in mammalian cells [23,24,25]. The connection between eIF5A and apoptosis induction is rather controversial. Indeed, suppression of eIF5A expression or inhibition of its hypusination induces apoptosis in various mammalian cells, whereas in a mouse diabetes model, reduction of eIF5A expression or inhibition of hypusine modification may cause suppression of apoptosis [25,26]. Moreover, in several mammalian cell types chemically-induced apoptosis is accompanied by increased levels of p53 as well as eIF5A [24]. In this work, we asked whether eIF5A is involved in controlling the expression of p53 at the translational level, and whether this influences apoptosis. We show that the inhibition of DHS enzyme activity by N1-guanyl-1, 7-diaminoheptane (GC7) induces an arrest of the eIF5A hypusination with a consequent arrest of UV-induced expression of p53. The effects of GC7 on p53 expression occur at the translational level since GC7 decreases 35S-Met metabolic labeling of neo-synthesized p53 and reduces the association of eIF5A to the ribosomes. Moreover, we show that inhibition of eIF5A hypusination induces a substantial decrease of protein synthesis under both normal and stress-induced growth cell conditions favoring the hypothesis that eIF5A controls the translation of many mRNAs. Lastly, decreased expression of p53 induced by GC7 counteracts the pro-apoptotic process triggered by UV treatment.

## 2. Results

### 2.1. Inhibition of Deoxyhypusine Synthase (DHS) Activity Affects p53 Stress-Induced Expression

The post-translational modification of eIF5A requires the enzymes deoxyhypusine synthase (DHS) and deoxyhypusine hydroxylase (DOHH), which change a specific lysine residue to hypusine [27]. This modification is essential for the function of the factor. To investigate the importance of eIF5A on the translational activity of mammalian cells under stress conditions, we used N1-guanyl-1, 7-diaminoheptane (GC7), a potent inhibitor of deoxyhypusine synthase. The study was performed on cell lines derived from human colon cancer where eIF5A is strongly up-regulated [28] and was shown to be involved in the regulation of apoptosis, and where p53 activity is well characterized [23,29]. In a first series of experiments we defined the optimal GC7 concentration causing an evident reduction of eIF5A hypusination levels. The post-translational modification of eIF5A was detected by immunoblotting using a specific polyclonal antibody [30]. As shown in Figure 1A,B, GC7 treatment for either 24 or 48 h resulted in a dose-dependent inhibition of eIF5A hypusination, keeping unaltered the total amount of protein. Successively, we focused our attention on the optimal conditions of UV-C irradiation inducing p53 expression in HCT-116 cells. We treated cells with increasing amount of UV ranging from 20 to 300 J/m^2^ for 24 h and the accumulation of p53 was compared by immunoblotting. As previously reported [31], the accumulation of p53 did not occur dose-dependently in UV-C irradiated cells. In fact, p53 expression increased at doses of UV ranging between 80 and 160 J/m^2^ while decreasing at 240 and 300 J/m^2^ (Figure 1C). We selected the dose of 80 J/m^2^ to monitor the expression of p53 in HCT-116 cells during time-course experiments. Specifically, p53 increased gradually, reaching its highest levels at 24 h post-irradiation (Figure 1D). eIF5A and its hypusinated form were not affected by UV-treatment, regardless of the exposure time.

Based on these preliminary results, our next experiments were performed irradiating HCT-116 cells with 80 J/m^2^ UV for 24 h and treating the cells with GC7 at a concentration of 80 µM for 48 h (schematic representation of the procedure in Appendix A). As demonstrated by trypan blue experiment (Appendix A), these conditions had no effect on cell viability using GC7 at 80 µM. The results revealed that p53 increased substantially after 24 h irradiation with UV-C, but treatment with GC7 inhibited p53 expression (Appendix A). Moreover, we note that, in contrast to what has been observed by other authors [23], the stress conditions adopted in all our experiments did not induce an evident variation of hypusinated and total eIF5A protein levels (Appendix A). To generalize the above findings to other stress conditions, HCT-116 cells were treated with doxorubicin 1 μM for 24 h, according to previous work [32], in the presence of GC7 80 μM. As illustrated in Figure 2B, GC7 caused a strong reduction of p53 induction, in agreement with the results obtained following UV treatment. Doxorubicin treatment did not affect the overall levels of eIF5A. We concluded this first series of experiments performing the same treatments on MCF-7 breast cancer cell lines, which also showed a reduction of UV-induced p53 expression after pre-treatment with GC7 (Figure 2D).

### 2.2. eIF5A Affects Stress-Induced p53 Expression 

The observations that treatment of HCT-116 cells with GC7 inhibited both eIF5A hypusination and UV-induced p53 expression does not demonstrate a relationship between the two events. Indeed, since GC7 is a spermidine analogue, it could affect the homeostasis of polyamines, possibly resulting in a non-eIF5A-specific inhibition of p53 expression [33]. In order to ascertain whether eIF5A specifically promoted the expression of p53 during UV-induced stress, we determined the levels of p53 in HCT-116 cells transiently transfected with a plasmid expressing eIF5A (pcDNA 3.1-eIF5A) and treated with UV. As shown by western blot analysis (Figure 3A), a significant increase in eIF5A expression was observed in cells transfected with pcDNA3.1-eIF5A compared to transfection with the empty plasmid. The over-expression of eIF5A was accompanied by increased levels of the hypusinated eIF5A form and promoted an enhanced expression of p53 protein after UV irradiation of HCT-116 cells (Figure 3A,B). Successively, to corroborate the proposed involvement of eIF5A in p53 expression, we examined the effects of siRNA-mediated suppression of eIF5A. As shown in Figure 3C, cells transfected with eIF5A siRNA showed a reduction of p53 expression, of the same order of magnitude as that observed upon treatment with GC7. Our findings are consistent with previous reports indicating that up-regulation and suppression of eIF5A increases and inhibits the levels of p53 protein, respectively [23,24,34]. To further confirm the involvement of hypusinated eIF5A on p53 expression, we also generated a construct expressing eIF5A bearing a single-point mutation of the conserved lysine at position 50 that was substituted with alanine (K50→A50) (pcDNA3.1-eIF5A(K50A). In this case, the increased expression of total eIF5A protein after transient transfection of HCT-116 cells with the mutated construct was not accompanied by a corresponding increase of the hypusinated form, as expected. Notably, over-expression of eIF5A (K50A) did not induce an increase of p53 protein levels (Figure 3D,E), similarly to what was observed by other investigators [23,24]. Overall, the results clearly indicate that eIF5A influences the expression of p53 under stress conditions with a critical role played by the hypusination modification. Similar results were found in previous reports [24]. 

### 2.3. GC7 Affects Global Protein Synthesis and Acts on p53 Expression at the Level of Translation

Recent information on eIF5A function suggest that the factor is required both to relieve ribosome stalling elicited by specific amino acid sequences in a nascent protein, such as poly-proline stretches, and to favor the translation termination phase at the level of stop codons [16,17]. To determine whether the altered expression of p53 in presence of GC7 was included in a more pervasive system of modified global de novo protein synthesis, we first determined the level of protein synthesis in HCT-116 cells treated with GC7. Initially, we adapted to our system the non-isotopic technique of Surface Sensing of Translation (SUnSET), that is an advantageous alternative to radioactive labeling [35]. Specifically, we measured the extent of global protein synthesis in control and UV and/or GC7-treated HCT-116 cells both at 24 and 48 h, evaluating puromycin incorporation into newly synthesized proteins by western blotting (Appendix A). As shown in Figure 4A, UV irradiation for 24 h inhibited, as expected, global protein synthesis, which was reduced to about 40–45% compared to the control. Addition of GC7 alone for 24 h also showed an inhibitory effect on translation, even if it was less pronounced than that obtained with UV stress. Not surprisingly, the combination of the two treatments decreased protein synthesis still further reducing the rate of protein synthesis to 30–35% respect to untreated cells, both at 24 and 48 h. Since it is well-known that p53 levels are largely regulated by changes in protein stability after DNA damage, we wished to ascertain whether GC7 influenced p53 half-life similarly to what was observed for other proteins such as HIF-1 [36]. To this aim, HCT-116 cells were pre-treated with the proteasome inhibitor MG132 before UV-irradiation in presence of GC7. As shown in Figure 4B, although the presence of MG132 slightly reduced p53 turnover (lane 2), GC7 still caused a marked reduction of p53 levels (lane 8). Interestingly, the mere presence of GC7 per se was not able to counteract the stabilization of p53 induced by MG132 (lane 6), suggesting that UV-stress increased the levels of p53 protein also through mechanisms other than its stabilization. Next, we set out to demonstrate that GC7 regulated p53 levels by influencing its translation. This was the most likely possibility, both because eIF5A is a translation factor and because several reports have suggested that translational regulation may also contribute to p53 induction after DNA damage [37,38]. As shown in Figure 4C, a brief exposure of HCT-116 cells to the protein synthesis inhibitor cycloheximide (CHX) caused a reduced induction of p53 after UV irradiation, indicating that, indeed, under the experimental conditions we adopted, the increase in p53 expression is dependent on de novo translation. To further prove this point, we labeled newly synthesized proteins with [35S]-Methionine and immunoprecipitated p53 under conditions where additional effects of protein turnover were avoided by pre-treating the cells with MG132. As shown in Fig. 4D, UV-exposure did elicit de novo translation of p53, which was, however, inhibited by treatment with GC7. The pattern was in agreement with the results observed incubating cells with puromycin (Figure 4A). Finally, to confirm that p53 is translationally regulated in response to DNA damage and that GC7 affects its synthesis at the translational level, we analyzed the levels of p53 mRNA associated with ribosomes following UV treatment. Briefly, after exposure of the cells to UV in presence/absence of GC7, cells were lysed, an aliquot was used for total cytoplasmic RNA isolation, the remaining fraction was ultracentrifuged to isolate ribosomes. RNA from the ribosome pellet, as well as that from the total cytoplasmic extracts, was purified and subjected to qRT-PCR. As shown in Figure 4E, p53 mRNA levels remained unvaried in UV treated cells in the presence of GC7, suggesting the drug influenced neither transcription nor RNA stability. However, in response to UV, the p53 mRNA relocated from the ribosome-free fraction to the ribosome-bound fraction while this relocation was inhibited in the presence of GC7. To confirm the specific control of p53 translation by eIF5A, we decided to analyze the behavior of SHMT1 mRNA, whose translation is induced by UV light [39] but which encodes a protein lacking an eIF5A-dependent motif. The results showed that UV treatment leads to an increased SHMT1 mRNA loading on the ribosomes compared to the control while the presence of GC7 in stress-induced cells does not affect SHMT1 mRNA ribosomal recruitment (Figure 4F).

### 2.4. GC7 Inhibits the Binding of eIF5A to the Ribosomes

Since the hypusinated status of eIF5A influences its capacity to bind ribosomes, we analyzed the distribution of the factor on the ribosome profiles of HCT-116 cells, before and after treatment with UV, GC7 or both. To this aim, whole cell extracts were subjected to velocity sedimentation on sucrose gradients, and the presence of hypusinated eIF5A or total IF5A in the various fractions was determined by western blotting. As shown in Figure 5A, eIF5A is very abundant in the cells. Under control conditions (upper left panel), it was found in most of the gradient fractions, from the pre-ribosomal ones to those containing monosomes and polysomes. Upon UV treatment (upper right panel), we observed an increased load of hypusinated eIF5A on the 80S fraction, while treatment with GC7 alone (bottom left panel) strongly inhibited, as expected, the association of eIF5A with 80S and polyribosome fractions. Upon treatment with both UV and GC7, binding of eIF5A to 80S ribosomes and polysomes was essentially abolished (bottom right panel). A quantification of eIF5A distribution on polysome profile (untreated and UV-treated cells) was performed (Appendix A).

The redistribution of eIF5A caused by the different treatments was accompanied by changes in the ribosomal profiles. Treatment with UV caused an increase of the 80S peak and a reduction of the polysomes. This was an expected effect, since it is known that stress conditions inhibit translation acting specifically on the initiation step through the phosphorylation of the α subunit of the translation initiation factor eIF2. Indeed, as shown in Figure 5B, UV irradiation increased the amount of Ser-51 phosphorylated eIF2. Interestingly, treatment with GC7 alone also caused an increase of 80S ribosomes and a decrease of the polysomal fraction, similar to what was observed by other authors [40]. However, in this case, we did not observe any increase of eIF2α phosphorylation, contrary to what reported by other researchers on other cell lines [41]. Therefore, GC7 may inhibit translational initiation acting independently of eIF2 or, more probably, the increase of the 80S peak is due to faulty termination and impaired ribosome recycling, in agreement to data showing that eIF5A affects translational termination [17,42]. Treatment with both UV and GC7 also reduced the polysomal fraction and increased the amount of monosomes.

As noted above, one of the effects of UV stress was to increase the ribosomal loading of eIF5A. Since it is difficult to quantify accurately the amount of a protein on gradient fractions, the issue was further analyzed by isolating ribosomes from HCT-116 cells subjected to the various treatments. As shown in Figure 5C, UV irradiation indeed stimulated the association of eIF5A with the ribosomes, while treatment with GC7 reduced it drastically. By contrast, no increase of eIF5A was observed in the post-ribosomal supernatant after UV irradiation. Overall, the data are consistent with the proposed role of eIF5A as translation elongation factor, and suggest that the function of the factor is especially important under stress conditions. 

### 2.5. Inhibition of p53 Expression by GC7 Reduces the Expression of Apoptosis Effectors Controled by p53

The p53 protein is a transcription factor that regulates different signaling pathways involved in the survival or death of the cells following stress conditions. These protective mechanisms include apoptosis, DNA repair and cell cycle arrest. Since post-translational modifications are important for p53 activation, we considered the possibility that GC7, in addition to affecting p53 translation, could influence the levels of phosphorylation of p53 on Ser15. To clarify this point, protein extracts of HCT-116 cells untreated and treated with GC7 and/or UV were analyzed by western blot with anti p53-Ser15 antibody. As shown in Figure 6A,B, the presence of GC7 did not affect the phosphorylation levels of p53-Ser 15. Next, since p21 is a well-known target of p53 and a recognized effector of the p53 activity [42], we assayed the levels of p21 mRNA in UV-irradiated HCT-116 cells in presence/absence of GC7. As shown in Figure 6C, p21 transcription increased following UV treatment, whereas it was inhibited by GC7. The observed transcriptional enhancement of p21 was p53-dependent, since it disappeared when the same experiment was performed using p53^−/−^ HCT-116 cells (Figure 6C). One of the most prominent physiological effects elicited by p53 after UV irradiation is the induction of apoptosis. Accordingly, we evaluated the activity of pro-apoptotic caspase 3 and 7. As shown in Figure 6D, UV-irradiated HCT-116 cells exhibited increased levels of caspase activity, while GC7 per se did not induce evident pro-apoptotic effects. However, the presence of GC7 in UV-irradiated cells inhibited the expression of pro-apoptotic caspases. The same was true for the pro-apoptotic member of the Bcl-2 protein family Bax, with a corresponding decrease of the levels of the anti-apoptotic factor Bcl-2 (Figure 6E). Therefore, inhibition of eIF5A activity appears to have a certain anti-apoptotic effect, presumably because of the decrease in p53 expression. On the score of these results, we next evaluated the effect of treatment with GC7 on the viability of the cells, using a colorimetric cell proliferation assay (CellTiter 96^®^ AQueous One Solution Cell Assay, Promega). As shown in Figure 6F, UV-irradiated cells had, as expected, a reduced viability compared to that of untreated cells, while treatment with GC7 alone had no similar effect. However, treatment with GC7 in UV-irradiated cells caused only a modest recovery of vitality, suggesting that the down-regulation of p53 expression caused by inhibition of eIF5A modification was unable to reverse the pro-apoptotic signaling triggered by UV-stress.

## 3. Discussion

eIF5A was initially proposed to be a factor implicated in translational initiation [14]. However, recent evidence indicates that hypusinated eIF5A promotes translation elongation and termination, acting on many different mRNAs and affecting global protein synthesis [15,16,43]. It is well established that eIF5A is a highly abundant and essential protein required for cell proliferation and survival of eukaryotic cells. Moreover, a strong correlation exists between increased levels of eIF5A and cancer [9]. Expression of the eIF5A gene has been examined in several cancers and cancer cell lines [10,13], both at the protein and RNA levels demonstrating unambiguously its over-expression in transformed cells. Other studies have also indicated a role for eIF5A in the regulation of apoptosis. 

In this work, we investigated the role played by eIF5A on the stress-induced expression of p53. In most experiments, eIF5A function was inhibited by using the chemical compound GC7, which interferes with hypusination. However, direct inhibition of eIF5A expression by siRNA was also tested. The results showed that hindering eIF5A function or expression consistently reduced p53 expression under conditions of both UV- and chemical (doxorubicin) stress. In most experiments, a colon cancer cell line (HCT-116) was used, but similar results were obtained using a different cell line (breast cancer cell line MCF-7). Our results are consistent with those of other authors showing that eIF5A is required for proper p53 expression in response to Actinomycin D [23]. Although the increase of p53 under stress conditions may be in part attributed to reduced protein turnover, there is also an important contribution of de novo translation [38,39,40,41,42,43,44]. Indeed, we observed that a brief exposure of HCT-116 cells to the protein synthesis inhibitor cycloheximide reduced the induction of p53 protein after UV irradiation. Furthermore, pulse-labeling with [35S]-Methionine showed a significant increase in metabolically labeled, newly synthesized p53 in the irradiated cells, thus demonstrating increased translation after DNA damage. Remarkably, the presence of GC7 decreased p53 labeling, showing unequivocally that eIF5A stimulates p53 expression at the level of protein synthesis. The importance of eIF5A for sustaining translation under stress conditions also stems from the fact that after UV-irradiation there was an increased loading of the factor on the ribosomes, which was inhibited by treatment with GC7. Strikingly, eIF5A was present on either monosomes and polyribosomes, confirming its primary role as a translation elongation factor. However, the treatment of the cells with only GC7 for 24 and 48 h lowered global protein synthesis at about 70% and 40% of wild-type rate, respectively. Hence, p53 is evidently only one of a set of mRNAs whose translation is significantly promoted by eIF5A. Data in the literature indicate that eIF5A is required to stimulate the translation of mRNAs, whose encoded proteins include not only polyproline stretches, but also other sequence motifs that may cause stalling of the ribosomes [17,43]. p53 mRNA represents an example of mRNA coding for poly-proline motifs [45]. Therefore, eIF5A could in principle influence its translation at the level of elongation preventing ribosome stalling at the level of these sites. That this is likely to be the case is shown, firstly, by the fact that GC7 did not induce an evident increase of eIF2-alpha phosphorylation, excluding a general translation block at the initiation level. Secondly, and more importantly, we show that the ribosomal loading of SHMT1 mRNA was not inhibited by GC7 despite being UV-induced like that of p53 mRNA [39]. Indeed, unlike p53 mRNA, SHMT1 mRNA lacks any evident motifs coding for poly-proline that may cause stalling of the ribosomes therefore requiring eIF5A to resume translation.

In summary, our data envisage a scenario where stress conditions induce increased eIF5A binding to the ribosomes, favoring the translation of a group of mRNAs including the one encoding p53. Inhibition of hypusination decreases the affinity of eIF5A for the ribosomes, causing a partial arrest of protein synthesis, which, however, affects only a subset of mRNAs, among which p53.

The translational inhibition of p53 appears to have some effect also on the apoptotic pathways. Indeed, treatment with GC7 concomitant with UV-irradiation reduced the expression of certain transcriptional targets of p53, such as the cyclin-dependent kinase (CDK) inhibitor p21 and members of the Bcl-2 family as Bax/Bcl-2. Additionally, a decreased activity of two well recognized apoptotic effector caspases as 3 and 7 [46] was detected. Therefore, at least in the cell line under analysis, treatment with GC7 exerted an anti-apoptotic effect, which was, however, rather weak and unable to fully counteract the loss of cell vitality caused by UV irradiation.

Understanding the precise mechanism whereby eIF5A can directly and differentially affect the translation of specific mRNAs, and which mRNAs are more responsive to variations in eIF5A levels and/or function, will require further experimental efforts. However, we do not encourage the idea that eIF5A selects actively the loading of specific mRNAs on the ribosomes during initiation, as proposed by other authors. Rather, we favor the idea that eIF5A promotes the translation of the mRNAs loaded on ribosomes under certain environmental conditions. For example, UV-irradiated cells show increased levels of p53 due in part to reduced protein turnover and in part to enhanced IRES-dependent translation. The role of eIF5A would be that of further stimulating translation avoiding the stalling of ribosomes at the level of the p53 polyproline tract or of other target sequences. This reasoning may apply to different cellular scenarios. For example, external stimuli enhancing cell migration could induce preferential translation of mRNAs whose protein products increase cell mobility. eIF5A could be a factor controlling cell migration if there are mRNAs encoding migration effectors of migration which contain sequence motifs where the ribosomes are prone to stall [46]. Regarding specifically p53, we could consider that many human tumors carry missense mutations in the p53 gene, that, in addition to abrogating the tumor-suppressor functions of wild-type p53, also endow the mutant protein with new activities that can contribute actively to tumor progression and to increased resistance to anticancer treatments. In these cases, it is conceivable that eIF5A could worsen the adverse effects induced by the gain-of-function p53 mutants, thereby behaving as an oncogenic factor. At the same time, it could sustain the translation of those mRNAs coding for proteins favoring cell transformation or tumor progression and containing stalling sites. 

Our results seemingly do not recommend the use of GC7 in anticancer therapy. However, extensive mutation searches showed that over 50% of human cancers carry loss-of-function mutations in the p53 gene [47,48]. In this genetic context, the use of GC7 obviously cannot affect the activity of p53; however, it could inhibit the translation of those cancer-promoting proteins containing ribosome-stalling motifs sensitive to the action of aIF5A. Furthermore, various lines of evidence indicate that, in addition to abrogating the tumor-suppressor functions of wild-type p53, the common types of cancer-associated p53 mutations also provide the mutant protein with new activities that may induce tumor progression and are commonly described as gain-of-function mutations [49]. In this genetic context, the inhibition of p53 expression would still be effective in anticancer therapy.

## 4. Materials and Methods

### 4.1. Cell Culture

The human colon cancer cell lines HCT-116 and the corresponding HCT-116 p53^−/−^ clone were kindly provided by Drs Leandro Castellano (Imperial College, London) and Claudia Carissimi (University of Rome “La Sapienza”). The cells were cultured in McCoy’s 5a Medium (Euroclone S.p.A. Pero (MI), Italy) supplemented with 10% fetal bovine serum (Thermo Fisher Scientific-Gibco, Rockford, IL, USA), 1% penicillin-streptomycin (Aurogene S.r.l. Rome, Italy) and 1 mmol/L L-glutamine (Gibco). MCF7 cells were kindly provided by Prof Rossella Maione (University of Rome “La Sapienza”) and maintained in Dulbecco’s modified Eagle’s medium (DMEM Gibco) supplemented with 10% fetal bovine serum (Gibco), 1% penicillin-streptomicin (Aurogene) and 1 mmol/L L-glutamine (Gibco). All cultures were grown in a humidified atmosphere at 37 °C with 5% carbon dioxide. Cultures were fed every 2–3 days and passaged when 70–80% confluent.

### 4.2. Treatment and Preparation of Cell Extracts

UV treatment: in experiments involving cell UV irradiation, HCT-116 cells at 50% confluence were washed twice with phosphate-buffered saline and then exposed to the indicated amount of UV-C (peak wavelength 275 nm) using the Stratagene Stratalinker UV Crosslinker1800. The medium was then replaced, and the cells were cultured under normal conditions for 24 h. GC7 treatment: DHS inhibition was performed adding GC7 (Merck Millipore Life Science S.r.l. Milan, Italy) to the cells as described in the text for 24 h or 48 h. Similar to other studies, no apparent cytotoxicity was noticed when cells were incubated with GC7 [50,51]. Cycloheximide (CHX) treatment: to observe the translational regulation of p53 expression after UV stress induction, HCT-116 cells were treated with CHX (50 μg/mL) (Merck KGaA-Sigma, Darmstadt, Germany) for 30 min before UV irradiation. Cells were analyzed 24 h after UV irradiation. Proteasome inhibitor MG-132 treatment: HCT-116 cells were grown for 24 h in presence of GC7 80 μM. Then, MG132 (Merck KGaA-Sigma) was added to the medium at a concentration of 10 µM for 1 h before UV-C treatment. Cells were left in the medium for the next 24 h in presence of GC7. Doxorubicin treatment: HCT-116 cells were grown for 24 h in presence of GC7 80 µM. Successively, doxorubicin (Merck KGaA-Sigma) was added to the cell medium at a concentration of 1 µM for the next 24 h.

### 4.3. Western Blotting Analysis

Cells were washed with phosphate-buffered saline (PBS) and incubated in RIPA buffer (50 mM Tris, pH 7.5, 150 mM NaCl, 0.1% Nonidet P-40, 1 mM phenylmethylsulfonyl fluoride, 1:100 dilution of Sigma protease inhibitor mixture). Cell lysates were subjected to Western blot analysis after removing the insoluble fraction by centrifugation at 13,000× *g* for 20 min. The protein concentration of all extracts was determined using the Bradford assay. In total, 20 µg protein samples were subject to SDS-PAGE, and transferred to nitrocellulose membrane (Amersham Protran-GE Healthcare, Little Chalfont, Buckinghamshire, UK). After blocking nonspecific binding of antibody with 5% non-fat milk, blots were probed with one of the following antibodies: p53 (FL-393; Santa Cruz Biotechnology Inc. Dallas, Texas, USA), GAPDH (FL-335; Santa Cruz Biotechnology), β-Tubulin (D-10; Santa Cruz Biotechnology), Anti-Puromycin (12D10; Merck KGaA), eIF2α (D7D3; Cell Signaling Technology, Inc. Danvers, MA, USA), Phospho-eIF2α (Ser51) (D9G8; Cell Signaling), Anti-Hypusine Antibody (Merck Life Science), Anti-eIF5A (ab32014; Abcam), Anti-RACK1 (BD), Ribosomal Protein L30 (G-12; Santa Cruz Biotechnology), Ser-15 p53 were kindly provided by Prof Claudio Talora (University of Rome “La Sapienza”). Primary antibodies were detected by binding horseradish peroxidase (HRP)-conjugated goat anti-rabbit IgG-HRP (sc-2004; Santa Cruz Biotechnology), goat anti-mouse IgG-HRP (sc-2005; Santa Cruz Biotechnology) and using an enhanced chemiluminescent visualization system (ECL Western Blotting Substrate, Thermo Scientific-Pierce Biotechnology, Rockford, IL, USA). Primary and secondary antibodies were diluted according to the manufacturer instructions. The images were captured by a BioRad ChemiDoc™ MP Imaging system (Bio-Rad, Hercules, California, USA).

### 4.4. RNA Isolation and qRT-PCR

Cells were collected and total RNA was extracted using Trizol reagent (Invitrogen, Carlsbad, CA, USA) following the manufacture’s protocol. cDNA was synthesized from 1 μg of total RNA using SensiFASTTM cDNA Synthesis Kit (Bioline, Meridian Life Science, Inc. Memphis, TN, USA). Quantitative real time PCR reactions were carried out using 5 μL of PowerUPTM SYBR^®^ Green Master Mix (Applied Biosystems by Life Technologies) and 4 μL of 1:4 cDNA. The analysis of amplification was performed by StepOneTM (Applied Biosystems by Life Technologies) denaturing DNA at 95 °C for 2 min, followed by 40 cycles at 95 °C for 3 s and 60 °C for 30 s. The following list shows the primers sequences used for qRT-PCR analysis of the specific targets: p53 (Forward 5’-GCGAGCACTGTCCAACAACA-3’ and Reverse 5’-GGATCTGAAGGGTGAAATA-3’), eIF5A (Forward 5’-GTCCCCAACATCAAAAGGAA-3’ and Reverse 5’-ACAGCTGCCTCCTCTGTCAT-3’), 18S rRNA (Forward 5′-TACCACATCCAAGGAAGGCAGCA-3′ and Reverse 5′-TGGAATTACCGCGGCTGCTGGCA-3′), p21 (Forward 5’-CTGTCTTGTACCCTTGTGCC-3’ and Reverse 5’-GAGTGGTAGAAATCTGTCATGCTG-3’), GAPDH (Forward 5’-AGCCACATCGCTGAGACA-3’and Reverse 5’-GCCCAATACGACCAAATCC-3’), Bax (Forward 5’-AGCAAACTGGTGCTCAAGG-3’ and Reverse 5’-TCTTGGATCCAGCCCAAC-3’), Bcl-2 (Forward 5’-AGTACCTGAACCGGCACCT-3’ and Reverse 5’-GCCGTACAGTTCCACAAAGG-3’), SHMT1 (Forward 5’-CAGGCTCCCCTGCAAA-3’ and Reverse 5’-GGGTTCACCTTGTAGGGCATA-3’). Melting curve analyzes were performed to verify the amplification specificity. Relative quantification of gene expression was performed according to the 2^–∆∆*C*t^ method using StepOneTM Software v2.3 (Applied Biosystems by Life Technologies) and normalizing to human housekeeping gene glyceraldehyde phosphate dehydrogenase (GAPDH) mRNA expression. All samples were performed in triplicate and each condition was repeated three times.

### 4.5. Plasmid and Transient Transfection 

Total RNA was extracted from HCT-116 human cell lines using Trizol reagent (Invitrogen, Carlsbad, CA, USA) following the manufacture’s protocol. It was reverse transcribed using SensiFASTTM cDNA Synthesis Kit (Bioline) and the human full-length eIF5A gene was amplified by Taq DNA Polymerase Hight Fidelity (Invitrogen) using the following primers (Sigma): Forward (5’-GTCCCCAACATCAAAAGG-AA-3’) and Reverse (5’-ACAGCTGCCTCCTCTGTCAT-3’). The resulting PCR product was cloned into pcDNATM 3.1(+) vector at the HindIII-XhoI sites and sequenced. The resulting cloning product was named pcDNA3.1-eIF5A. This construct was used as mutagenesis template to generate the mutant pcDNA3.1-eIF5A (K50A) by Q5^®^ Site-Directed Mutagenesis Kit (NewEngland Biolabs Ltd., Ipswich, MA, USA) according to instructions of the supplier. The primers used for site-directed mutagenesis were: Forward eIF5A K50 5′- GAAGACTGGCGCGCACGGCCAC–3′ and Reverse eIF5A K50 5′- GAAGTAGACATCTCGACG–3′. The positive mutants were selected by DNA sequencing. The wild type and the mutant K50A of eIF5A were transiently transfected by Lipofectamine LTX kit (TermoFisher), as indicated by the supplier. HCT-116 cells, grown in McCoy’s medium supplemented with 10% FBS and L-Glu 2%, were seeded in 24-well plates at a density of 3 × 105 cells per well. After 24 h, cells were transfected with the expression vector containing human eIF5A sequence. The control corresponded to the empty pcDNA3.1 vector and it was added to each sample ensuring an equal amount of total DNA. Cells were lysed at the times described in the Results and the levels of eIF5A protein were detected by immunoblotting.

### 4.6. RNA Interference and Transfection 

Small interfering RNAs (siRNAs) against eIF5A were obtained from QIAGEN (Hilden, Germany) (siRNA #1,2), as already described from other authors (26). The following eIF5A and control siRNA were used for silencing experiments: si-eIF5A # 1 5’- UUCGCGCGAGUUGGAAUCGAA-3’, si-eIF5A # 2 5’- AAAGCGGUGGAUUCUGGCAAA-3’ and si-CTRL 5’- AGCUUCAUAAGGCGCAUGC-3’. All transfections were carried out using Lipofectamine 2000 Transfection Reagent (Thermo Fisher Scientific) according to manufacturer’s instructions.

### 4.7. Polysomal Profiles

HCT-116 cells (about 5 × 10^6^ cells) were treated with CHX to a final concentration of 100 μg/mL for 15 min at 37 °C. After washing the monolayer once with ice-cold PBS 1X + CHX (50 μg/mL), the cells were lysed in 500 µL of ice-cold lysis buffer (10 mM Tris-HCl pH 7.4, 10 mM KCl, 15 mM MgCl_2_, 1 mM DTT, 1% Triton-X 100, 1% DOC, 100 μg/mL CHX, heparin 1 mg/mL, 200 U/mL SUPERase·In and protease inhibitor cocktail) on ice. Cell debris were removed centrifuging for 10 min at 10,000× *g* at 4 °C. 10 OD260 units of supernatant was layered on top of a linear 15–50% (*w*/*v*) sucrose gradient containing 10 mM Tris-HCl pH 7.4, 5 mM MgCl_2_ and 10 mM KCl, 0.5 mM DTT and 0.1 mg/mL CHX. The gradients were centrifuged at 4 °C in a SW41 Beckman rotor for 3 h at 39.000 rpm and unloaded while monitoring absorbance at 254 nm with ECONO UV monitor instrument (Biorad). Fractions (0.5 mL) were collected in 18 tubes and precipitated with TCA and 1/100 volume of 2% DOC. The dried pellet was resuspended in SDS sample buffer 1X and analyzed by western blotting.

### 4.8. Ribosomes Purification

The HCT-116 cells were treated with UV and GC7 as previously described. At the end of treatments, the cells were lysed in 500 µL of lysis buffer (20 mM Tris/HCl pH 7.4, 10 mM KCl, 10 mM MgCl_2_, 2 mM EDTA, 2 mM DTT). The lysates were left on ice for 30 min and then centrifuged at 12,000× *g* for 10 min. The supernatants were layered on top of 30% sucrose cushion containing 20 mM Tris/HCl pH 7.4, 10 mM KCl, 10 mM MgCl_2_, 2 mM EDTA, 2 mM DTT and centrifuged at 40,000× *g* for 18 h. The supernatant was collected (post 80S) and as well the ribosomes pellet (80S). Ribosomes pellet was resuspended in 100 μL of lysis buffer and quantified at λ = 260 nm. Equimolar amounts of 80S were analyzed for eIF5A content by western blot analysis.

The analysis of p53 mRNA bound to the ribosomes was performed analyzing equal amounts of total RNA extracted from both 80S pellets and whole cell extract. RT-qPCR was carried out according as described above. The analysis of SHMT1 mRNA bound to the ribosomes was performed analyzing equal total RNA amount extracted by both 80S pellets and fraction ribosomes free. RT-qPCR was carried out according to what described above.

### 4.9. Metabolic Labeling and Immunoprecipitation

For metabolic labeling, HCT-116 cells were pre-treated with proteasome inhibitor MG132 (Sigma) 10 μM for 1 h and then irradiated with UV-C 80 J/m^2^. After 24 h, cells were preincubated in DMEM without methionine and cysteine with 5% dialyzed FBS (Gibco by Life Technologies) for 1 h and labeled with 100 μCi/mL of [35S]-Methionine (Amersham) for 15 min. For immunoprecipitations, cells were washed with PBS and lysed in RIPA buffer. The insoluble fraction was removed from the cell lysate by centrifugation at 13,000× *g* for 20 min. Lysates were pre-cleaned by incubation with protein A SepharoseTM CL-4B (Amersham) and rabbit IgG (Santa Cruz Biotechnology). Precleared lysates were incubated with rabbit polyclonal anti-p53 antibody (FL-393; Santa Cruz Biotechnology) overnight. The p53-antibody complex was recovered by binding to protein A SepharoseTM CL-4B (Amersham), and the beads were washed once with RIPA lysis buffer and twice with PBS 1X. Immunoprecipitated p53 was subjected to gel electrophoresis, then transferred to nitrocellulose membrane. The amount of labeled p53 on the membrane was estimated by exposure to X-ray film, and the total amount of immunoprecipitated p53 on the membrane was detected by Western analysis with mouse monoclonal anti-p53 antibody (DO-1; Santa Cruz Biotechnology).

### 4.10. SUnSET

The HCT-116 cells were treated with UV and GC7 as previously described and 20 h before the end of the treatment puromycin was added to growing cells to a final concentration of 2 μM. After 20 h, the cells were lysed with RIPA buffer and the protein concentration of extracts was determined using the Bradford assay. To detect the truncated proteins that arise from puromycin’s misincorporation into polypeptides, 20 μg of protein were loaded onto a 15% SDS-PAGE gel. Proteins were transferred on nitrocellulose membranes. Prior to immunoblotting, membranes were stained with Ponceau Red to ensure an equal loading of proteins. Membranes were incubated with anti-puromycin antibodies. Protein synthesis was estimated based on the intensity of immunoreactive bands using BioRad ChemiDoc™ MP Imaging system.

### 4.11. Cell Proliferation Assay (MTS)

The number of viable cells in proliferation or cytotoxicity assays was determined using the CellTiter 96^®^ AQueous One Solution Cell Proliferation Assay (Promega). The MTS tetrazolium compound (Owen’s reagent) is bio reduced by cells into a colored formazan product that is soluble in tissue culture medium. The quantity of formazan product as measured by absorbance at 490 nm is directly proportional to the number of living cells in culture. Assays were performed by adding 20 µL of CellTiter 96^®^ AQueous One Solution Reagent (Promega) into each well of the 96-well assay plate containing the samples in 100 µL of culture medium. We incubated the plate at 37 °C for 4 h in a humidified, 5% CO_2_ atmosphere and then recorded the absorbance at 490 nm.

### 4.12. Apoptosis Assay

Following the treatments described above, cells were subjected to Caspase 3/7 activities measurement with Caspase-Glo assay kit (#G8091, Promega, Madison, WI, USA) according to the manufacturer’s instructions. Briefly, assays were performed into 96-well plate containing 10,000 cells/well. After UV and/or GC7 treatment, the plates were removed from the incubator and allowed to equilibrate to room temperature for 30 min. In total, 100 μL of Caspase-Glo reagent was added to each well; the content of the well was gently mixed with a plate shaker at 300–500 rpm for 30 s. The plates were then incubated at room temperature for 1 h. The luminescence emission of each sample was measured using a TriathlerTM multilabel tester (Hidex, Turku, Finland). The experiments were performed in triplicate and repeated at least three times.

### 4.13. Trypan Blue Exclusion

HCT116 cells were treated with GC7 according to the protocol above. The cells, 24 and 48 h after GC7 exposure, were harvested by trypsinization, pelleted by centrifugation and then resuspended in serum-free-McCoy’s 5a. The suspension was then diluted 1:1 with 0.4% trypan blue (Invitrogen), and the viable and non-viable cells were counted using a haemocytometer according to the following formula: % viable cells = [1.00 − (Number of blue cells/Number of total cells)] × 100.

### 4.14. Statistical Analysis

Data were expressed as mean ± standard deviation (SD). Differences were tested by two-tailed *t*-test. Values with *p* < 0.05 were considered statistically significant.

## Figures and Tables

**Figure 1 ijms-21-04583-f001:**
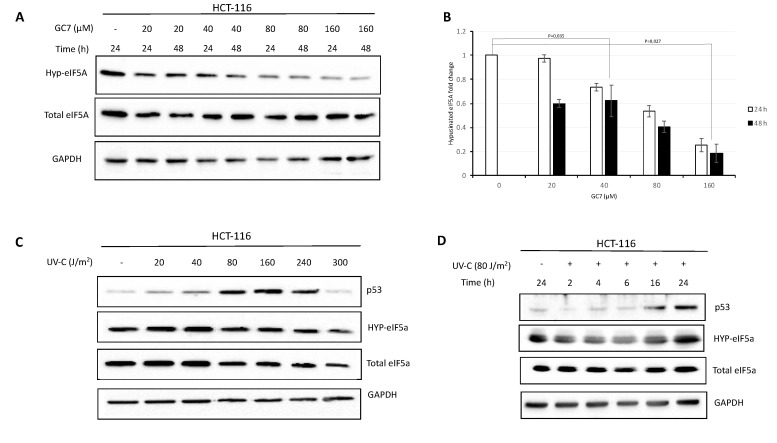
Dose dependence and time course of the hypusinated eIF5A and p53 protein expression in response to GC7 and UV cell treatment. (**A**) HCT-116 cells were treated for 24 and 48 h with different concentrations of GC7 as indicated. Expression of hypusinated eIF5A was detected by western blot analysis using a specific polyclonal antibody. (**B**) Intensity of bands was detected by the Bio-rad Image Lab Software 5.2.1 and the signal of interest was normalized to control GAPDH. The most intensive signal was set as 1. Data are means ± SD (*n* = 4). *p* values reported to the upper side of the histograms were calculated versus control group. (**C**) HCT-116 cells were treated with the indicated doses of UV-C for 24 h. Whole cell extracts were prepared and analyzed by immunoblotting with the antibodies showed in the figure. (**D**) Kinetics of p53 accumulation were performed treating HCT-116 cells with 80 J/m^2^ UV-C radiation for the times shown. Whole cell extracts were prepared and analyzed by immunoblotting with the indicated antibodies. A representative image of at least three independent experiments is shown for each analysis.

**Figure 2 ijms-21-04583-f002:**
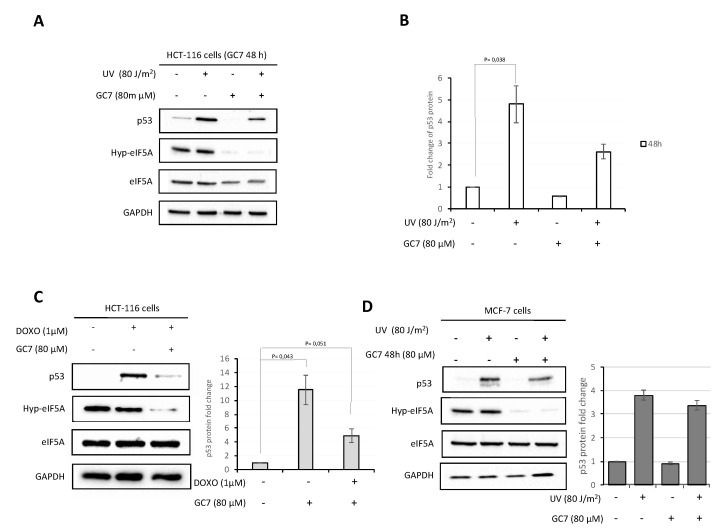
GC7 has an inhibitory effect on p53 expression. (**A**) The HCT-116 cells were treated with GC7 at a concentration of 80 μM for 48 h and subject at UV-C irradiation where indicated. Cell lysates were analyzed by immunoblotting with the antibodies showed in the figure. (**B**) The p53 expression level was normalized to GAPDH by the Bio-rad Image Lab Software 5.2.1 and the value of p53 in untreated cells was set as 1. Data are means ± SD (*n* = 5). *p* values reported to the upper side of the histograms were calculated versus control group. (**C**) Treatment of HCT-116 cells with GC7 at a concentration of 80 μM 48 h inhibited p53 expression induced by doxorubicin 1 μM. Whole cell extracts were prepared and analyzed by immunoblotting with the indicated antibodies. Intensity of bands was detected by the Bio-rad Image Lab Software 5.2.1 and the signal of interest was normalized to control GAPDH. The value of p53 in untreated cells was set as 1. Data are means ± SD (*n* = 4). *p* values reported to the upper side of the histograms were calculated versus control group. (**D**) Treatment of MCF-7 breast cancer cell lines with GC7 (80 μM) for 48 h and UV-C irradiation at the indicated dose. Whole cell extracts were prepared and analyzed by immunoblotting with the indicated antibodies. Each image is representative of at least three independent experiments.

**Figure 3 ijms-21-04583-f003:**
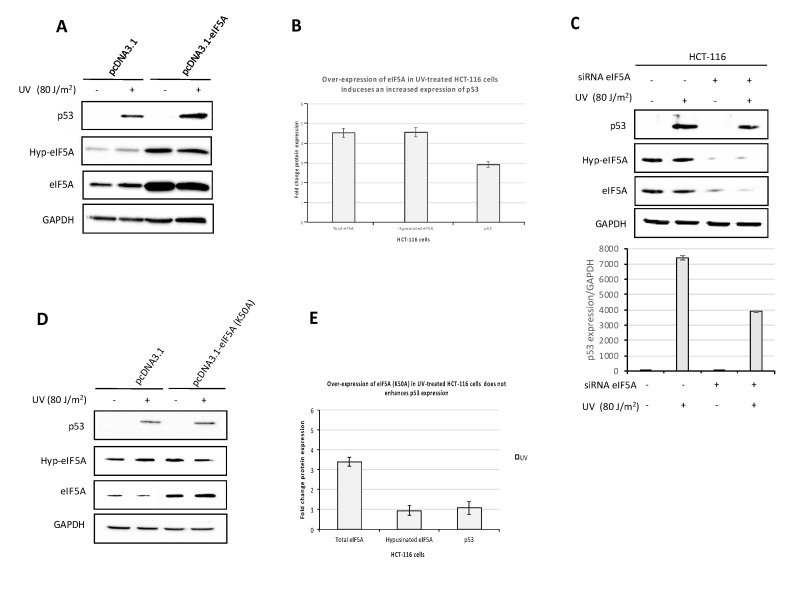
eIF5A regulates p53 protein expression. (**A**) HCT-116 cells were transiently transfected with the pcDNA 3.1- eIF5A expression vector and the control plasmid pcDNA3.1, respectively. Twenty-four hours after transfection, cells were UV-C irradiated and were then grown for another 24 h. Cell lysates were analyzed by immunoblotting with the indicated antibodies. The expression level was normalized to GAPDH. (**B**) The intensity of bands was analyzed by the Bio-rad Image Lab Software 5.2.1 and the values of the histogram represent the fold change expression of indicated proteins in UV treated cells comparing their expression in HCT-116 cells transfected with the specific constructs with respect the empty one. Data are means ± SD (*n* = 3) (**C**) HCT-116 cells were transfected with siRNA targeting eIF5A mRNA 24 h before UV-C irradiation and were then grown for another 24 h. Cell lysates were analyzed by immunoblotting with antibodies to p53, hypusinated form of eIF5A and total eIF5A. Histogram represents p53 expression level normalized to GAPDH. (**D**) Over-expression of the construct pcDNA3.1-eIF5A(K50A) containing single-point mutation and UV-C irradiation of the cells. Cell lysates were analyzed by immunoblotting with antibodies to p53, hypusinated form of eIF5A and total eIF5A. The expression level was normalized to GAPDH. (**E**) The intensity of bands was analyzed by the Bio-rad Image Lab Software 5.2.1 and the values of the histogram represent the fold change expression of indicated proteins in UV treated cells comparing their expression in HCT-116 cells transfected with the specific constructs with respect the empty one. Data are means ± SD (*n* = 3).

**Figure 4 ijms-21-04583-f004:**
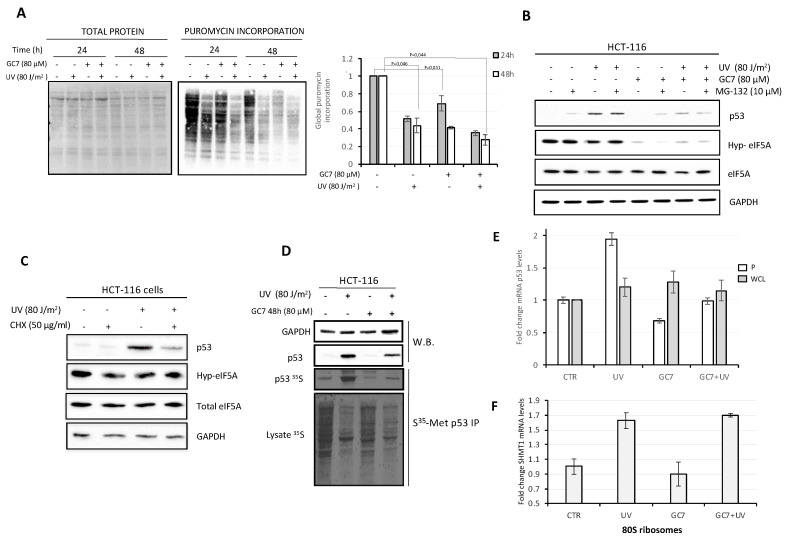
GC7 inhibits global protein synthesis and perturbs p53 expression at the level of translation. (**A**) SUnSET assay quantified by histogram on the right. The bar graph summarizes the level of total puromycin proteins measured by densitometry on immunoblottings of three independent experiments and the value of puromycin signal in untreated cells was set as 1. On the left, membrane was stained with Ponceau S solution prior to immunoblotting to verify equal loading of total protein in all lanes. (**B**) HCT-116 cells were incubated with 10 μM of proteasome inhibitor MG132 for 1 h before to be UV-C irradiated at the indicated dose. When indicated, HCT-116 cells were treated with GC7 for 48 h at the indicated dose (schematic presentation of the experiment in Appendix AB). Whole cell extracts were prepared and analyzed by immunoblotting with the indicated antibodies. (**C**) HCT-116 cells were treated with 50 μg/mL CHX 30 min before UV-C irradiation at the indicated dose. Cells were analyzed 24 h after UV irradiation assessing the levels of indicated proteins by immunoblotting. The expression level was normalized to GAPDH. (**D**) p53 protein was immunoprecipitated from HCT-116 cells that was labeled with [35S]-methionine 24 h after exposure to 80 J/m^2^ UV-C and was assessed by autoradiography. Whenever indicated, cells were treated with GC7 24 h before UV-C irradiation. Cells were pre-treated with 10 μM MG132 for 1 h before UV-C irradiation. Immunoblotting showed amount of GAPDH and p53. GAPDH used as loading control. Analysis of whole cell extracts showed decreased amounts of [35S]-methionine incorporated into the irradiated and/or GC7 treated cells (bottom panel). (**E**) The histograms show the levels of p53 mRNA in ribosome pellet (P) and in whole cell lysate (WCL) fractions of HCT-116 cells treated as described. The bar graphs represent the relative fold changes of target mRNA relative to that of GAPDH and of 18S rRNA for whole cell lysate and ribosomes, respectively. The results are the average of three independent experiments. (**F**) The histograms show the levels of SHMT1 mRNA in ribosome pellet (80S) of HCT-116 cells treated as described. The bar graphs represent the relative fold changes of target mRNA relative to that of 18S rRNA. The results are the average of three independent experiments.

**Figure 5 ijms-21-04583-f005:**
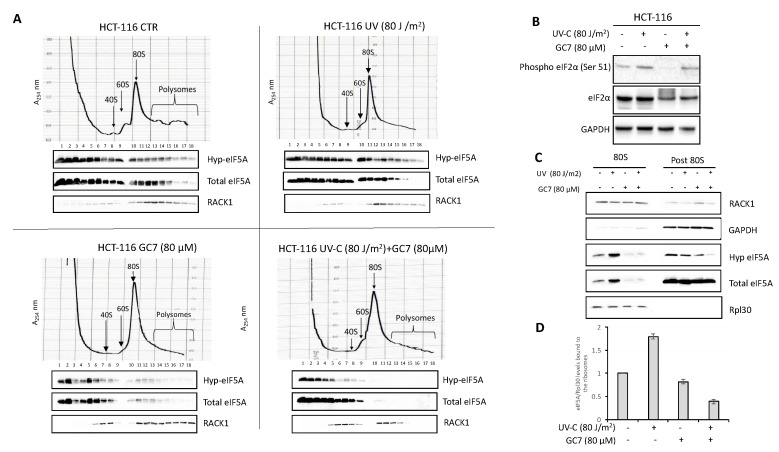
GC7 and UV treatments affect the binding of eIF5A to the ribosomes of HCT-116 cells. (**A**) HCT-116 cells polysomes profile was analyzed by density gradient centrifugation. After the indicated treatments, 10 ODs of the cell lysates were loaded onto a sucrose gradient and fractionated. Optical scans (OD254 nm) of the gradients are shown. Fractions were analyzed by immunoblotting with the indicated antibodies. (**B**) HCT-116 cells were treated with the indicated doses of UV-C and GC7. Whole cell extracts were prepared and phosphorylation and protein levels were determined by immunoblotting with the appropriate antibodies, as indicated. (**C**) Ribosomes (80S) were purified from HCT-116 cells by sedimentation through a sucrose cushion and were probed with the indicated antibodies. The post 80S represents the supernatant after ribosomes precipitation. The absence of Rpl30 protein shows the ribosome-free fraction. (**D**) Intensity of bands was detected by the Bio-rad Image Lab Software 5.2.1 and the signal of interest was normalized to control Rpl30. The value of eIF5A in untreated cells was set as 1. Data are ± SD (*n* = 3).

**Figure 6 ijms-21-04583-f006:**
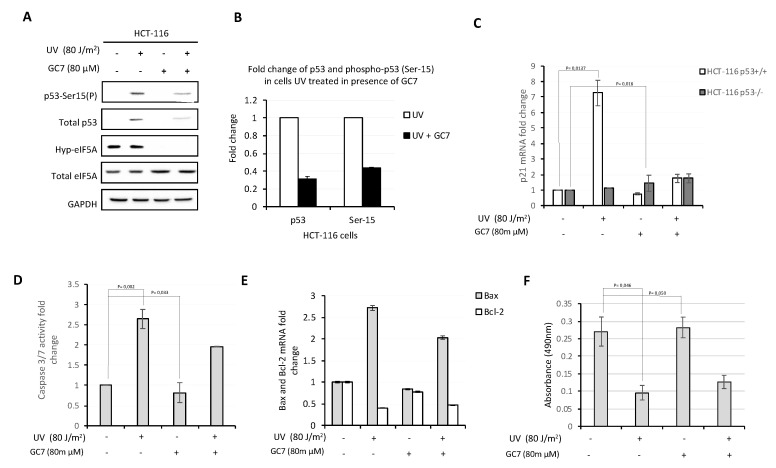
GC7 counteracts pro-apoptotic effects of p53. (**A**) Analysis of p53 phosphorylation levels on Ser15 in HCT-116 cells treated with UV and/or GC7 performed by immunoblotting with the indicated antibodies. (**B**) Intensity of bands was detected by the Bio-rad Image Lab Software 5.2.1 and the signal of interest was normalized to control GAPDH. The values of p53 and p53-Ser15(P) in untreated cells was set as 1 and correspond to the control. The intensity of the signals in treated cells was calculated versus control group and reported as their ratio. Data are means ± SD (*n* = 4). (**C**) qPCR analysis of p21 mRNA expression in HCT-116 and HCT-116 p53^−/−^ cells after UV and GC7 treatment. Results are presented in terms of a fold change after normalizing with GAPDH mRNA. Each value represents the mean ± S.D. of three independent experiments. *p* values reported to the upper side of the histograms were calculated versus control group. (**D**) Caspase 3/7 activity was measured in HCT-116 cells after UV and/or GC7 treatment as indicated. Each value represents the mean ± S.D. of three independent experiments. The value of Caspase 3/7 activity in untreated cells was set as 1. *p* values reported to the upper side of the histograms were calculated versus control group. (**E**) qPCR analysis of Bax and Bcl-2 mRNA in HCT-116 cells treated as indicated. Each value represents the mean ± S.D. of three independent experiments. Results are presented in terms of a fold change after normalizing with GAPDH mRNA. The value of target mRNAs in untreated cells was set as 1. (**F**) The viability of HCT-116 cells after UV and/or GC7 treatment was evaluated via MTS assay. Each value represents the mean ± S.D. of three independent experiments and *p* values reported to the upper side of the histograms were calculated versus control group.

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
