# Peer review of "Inhibition of Eukaryotic Translation Initiation Factor 5A (eIF5A) Hypusination Suppress p53 Translation and Alters the Association of eIF5A to the Ribosomes"

_ijms, 2020, doi:10.3390/ijms21134583_

Round 1

Reviewer 1 Report

The manuscript “Inhibition of eIF5A hypusination suppress p53 translation and alters the association of eIF5A to the ribosomes” by Martella et al, highlights the mechanism of hypusination mediated regulation of eIF5A action. The presented results stand to serve as an important template for future investigations and translational science and the authors have done an excellent work for it. I request the authors to address the following:

  1. Pg-2-line-63: “…cells..” à “…cell..”
  2. Do the authors envision anti-apoptosis effect of GC7 to interfere in anti-cancer use? Please include in Discussion
  3. Just curious: I was wondering if the authors might know how might eIF5A inhibition affect a viral infection like the SARS-CoV2?

Reviewer 2 Report

Well-written paper about significance of eIF5a translation factor for p53 synthesis. All experiments are clear and well-presented. Conclusions made are logical and consistent with data obtained. I have only minor questions and remarks.
1. Why authors did not present quantification of western blot at figure 3C? It seems that p53 level at eIF5A knokdown is not so differs from level in control UV-irradiated cells.
2. Loading controls at figures 4A and 4D are missing.
3. Figure 5A - difference in redistribution of eIF5a at ribosomes from unirradiated and irradiated is not well distinguishable. May be, quantification is needed.
4. Please, check hyperlinks and references at lanes 40, 56, 428.

Reviewer 3 Report

  1. This manuscripts needs language editing.
  2. For Fig 1 A: Usually, hyp-eIF5A is normalized to total eIF5A. Even though the actin looks similar but the total eIF5A 24h without GC7 treatment looks higher than the treatment. Please change it with representative band.
  3. 1C and 1D, author observed the effect of UV treatment with P53 expression. But author also add eIF5A data in the figure but author did not mention in the result part what is the purpose of eIF5A data in the figure.
  4. Please change the title for Fig1. It is easier for reader to understand what is the purpose of Fig1, the current topic did not give much information.
  5. This study is somewhat confusing. Author did not show the clear pathway and design of study need to be improved. This study would be better if author add animal study. 
  6. Why author choose caspase 3/7? The most common caspase for evaluating apoptosis is caspase 3 and caspase 1. Please add the gel data for caspase data, bax and bcl2. Apoptosis are very broad and author only observe caspase 3/7, bax and bcl2. 
  7. Discussion needs to be improved. Author needs to discuss all the findings in the discussion

Round 2

Reviewer 3 Report

The data in this study is not sufficient for publication yet, such as comment in point 6.